# Flower Position and Clonal Integration Drive Intra-Individual Floral Trait Variation in Water-Hyacinth (*Eichhornia crassipes*, Pontederiaceae)

**DOI:** 10.3390/biology14020114

**Published:** 2025-01-23

**Authors:** Guilherme Ramos Demetrio, Luziene Seixas, Flávia de Freitas Coelho

**Affiliations:** 1Plant Ecology Lab, Penedo Educational Unit, Campus Arapiraca, Federal University of Alagoas, Av. Beira Rio, s/n, Centro Histórico, Penedo 57200-000, AL, Brazil; luzieneseixas@hotmail.com; 2Graduate Program in Ecology (PPG-Ecology), State University of Campinas, Campinas 13083-862, SP, Brazil; 3Departamento de Biologia, Universidade Federal de Lavras, Campus Universitário, Lavras 37200-000, MG, Brazil; flaviafcoelho@gmail.com

**Keywords:** clonality, environmental heterogeneity, flower morphology, sexual reproduction, stressing conditions

## Abstract

In this study, we explored how the position of flowers and the connection between plant parts affect the development of individual flowers in water hyacinth (*Eichhornia crassipes*). This plant is known for its invasive ability and unique clonal growth, where new plants form from parts of the parent. We aimed to understand if this clonal connection helps stabilize flower traits when resources are limited. We found that flowers at the bottom of the plant receive more resources, making them larger and more attractive for pollinators. When this clonal connection was removed or the plants were stressed, flower size and pollinator-attracting traits decreased, especially in flowers located higher on the plant. These findings suggest that clonal integration helps water hyacinths maintain reproductive traits even in challenging conditions, possibly aiding their spread. This research offers insights into how plants adapt to varying environments, which could help in managing invasive species like water hyacinth.

## 1. Introduction

Ecological and evolutionary studies traditionally emphasize variability among populations or specific functional traits [1,2,3]. However, individual-level differences have gained attention and are increasingly incorporated into ecological studies [4,5]. Intraspecific trait variability (ITV) is crucial to evolutionary processes and is a key process to be considered when the focus is the development of a species-centered study [6], as natural selection favors resource allocation that maximizes plant fitness [7,8]. ITV plays a key role in individual and population performances [9]. This intraspecific variation is particularly important for plants because it is ubiquitous in these organisms, and many traits that vary at this scale have significant impacts on plant fitness [10].

Traits like flower size and patterns of sexual investment within individuals significantly influence plant mating and are thus subject to natural and sexual selection [11,12,13]. Within-plant variation can affect interaction patterns [14], with much of this variation concentrated in flowers and inflorescences compared to inter-individual levels [15,16,17]. It is necessary to understand this high level of variation when one recognizes that the flowers are not a unique organ, but a group of several independent organs covarying [18,19,20] and exhibiting high degrees of integration in working together as reproductive structures. Variations in floral traits within individuals or inflorescences can reflect environmental or ecological factors, such as plant size [21,22], pollinator behavior, out-crossing rates [23], and local resource availability [24]. These differences can also be explained by architectural effects related to flower position [25] that interact with resource availability for flowers at different positions [26,27].

Flower position strongly influences the expression of floral traits by shaping resource allocation patterns within inflorescences [28]. The internal distribution of resources, driven by the proximal-to-distal reduction in vascular size [29,30], is particularly evident in plants with acropetal flowering patterns [30,31,32], such as *Eichhornia crassipes*, and may lead to the formation of a resource gradient. This gradient may result in differential resource distribution, where flowers at the base of the inflorescence, typically among the first to develop and reach anthesis, receive more resources than those positioned distally [33]. These basal flowers are positioned to act as primary sinks, drawing resources from source organs, such as leaves, as they are translocated. This aligns with the developmental priority hypothesis, which suggests that proximal flowers attract more resources due to their early initiation and establishment as active sinks. In this way, resources are strategically allocated to maximize reproductive success and environment-developmental coordination [34], ensuring that basal flowers, which are more likely to set fruit due to their proximity to the vascular supply, receive preferential investment. However, there is still a lack of information on how flower position along a plant axis affects floral traits and their coordination [35]. In this sense, plants in different resource conditions can present different patterns of intra-individual variation. For example, different resource levels, such as different nutrient conditions, are related to intra-individual differences in the flowers of *Pyrus bourgaeana* [36].

Clonal plants are species that reproduce asexually through vegetative structures, such as runners, stolons, or rhizomes, allowing them to form genetically identical offspring or clones, which, in most cases, remain attached to their parental plants [37]. In these species, clonal integration allows resource sharing between connected ramets, stabilizing growth and reproduction in heterogeneous environments. Recent studies have shown that integration helps clonal plants adjust to variable resource conditions by facilitating the translocation of water, nutrients, and photosynthates among ramets [38]. For instance, Cao et al. [39] discussed how clonal plants, such as *Hydrocotyle vulgaris* and *Duchesnea indica*, adapt resource acquisition strategies based on environmental heterogeneity through physiological integration, which may stabilize resource availability across connected ramets. Thus, clonal integration is a key mechanism for resource exploitation, allowing nutrient and photosynthate exchange between parental and daughter ramets, with functional differentiation among ramets (refs. [40,41,42]), which may influence the intra-individual floral trait variation patterns. Especially in the case of invasive clonal species, such as *Eichhornia crassipes*, clonal integration may provide a competitive advantage, enabling them to exploit resource-rich areas more effectively than native species [43]. This trait may contribute to the success of *Eichhornia crassipes* in diverse environments by stabilizing resource availability and supporting trait consistency across ramets.

We aimed to examine how clonal integration, resource availability, and flower position interact to determine intra-individual variation in sexual reproductive traits in *E. crassipes*. We hypothesized that (i) ramets that remain attached to parental plants should show lower floral variation due to higher availability of resources when compared to defoliated and isolated ramets and, (ii) defoliated ramets should present the greatest influence of floral position on sexual reproductive traits, as flowers in more proximal positions on the inflorescence would act as a strong resource drain [44,45,46].

## 2. Materials and Methods

### 2.1. Study Species

*Eichhornia crassipes* (Mart.) Sölms. is a free-floating aquatic macrophyte [47,48,49], native to the Amazon river basin [49]. It has been recognized as an aggressive invasive species [50], reaching a worldwide distribution [51]. This increase in its geographic range may be related to the vigorous reproductive processes that occur both via asexual and sexual pathways [47,52]. Sexual reproduction is not constrained by oligotrophic habitats [53] and does not seem to be a weakening factor for asexual reproduction [52]. In fact, clonal integration has been highlighted as a key driver of sexual reproductive success in this species, enhancing its success [54]. Sexual reproductive structures are grouped in an inflorescence that generally arises from the apical meristem, bearing showy light purple flowers that open in an acropetal order (from bottom to top of the inflorescence), as can be seen in Figure 1.

### 2.2. Plant Sampling

We collected plant material from a monospecific mat, consisting of a large, continuous patch of clonal plants composed of a single species (*E. crassipes* in this study), in which vegetative offspring (e.g., generated by runners, rhizomes, or stolons) spread from a central parent plant to form a dense network of genetically identical individuals. This mat was located at Represa do Funil, Ijaci, Lavras, Minas Gerais, Brazil. We sampled adult ramets (identified by the presence of newly produced sexual reproductive structures or their remains, as old floral scapes) with no signs of foliar herbivory or diseases. We collected a total of 90 ramets that were put in plastic bags filled with some water to avoid root desiccation and took them to a greenhouse at the Federal University of Lavras in order to carry out the experiment.

### 2.3. Greenhouse Experiment

We distributed the ramets among 18 pots (Plasútil, Bauru, Brazil) (5 ramets/pot) filled with 17 L of tap water. The ramets were kept in the greenhouse for a two-week acclimation period without interference. After this time, we selected 18 similarly sized (~3 leaves and 10 cm of diameter) ramets and placed each one in individual pots containing 17 L of tap water. The ramets were cultivated until the production of asexual offspring for each of the selected ones (~1 week). This first generation of ramets produced under greenhouse conditions is referred to as “parental” hereafter. Parental ramets remained attached to the ramets from which they derived until they also produced a generation of asexual offspring (Figure 2). 

After this, parental ramets and their offspring were used as the basic experimental unit. These groups of parental ramets and their first asexual offspring were placed in 27 pots filled with 17 L of tap water, one group per pot. After this, we set three treatments (n = 9 groups per treatment) on these parental-daughter ramets groups. The first treatment, ‘clonal treatment’, consisted of the maintenance of nine groups as originally conceived, with the daughter ramet attached to the mother ramet. The second one, ‘Isolation’, involved nine daughter ramets separated from their parent ramets. The third treatment, ‘Defoliation and Isolation’, involved nine daughter ramets severed from their parent plants and completely defoliated at the experiment’s start to simulate resource limitation (Figure 2). The experiment was carried out over three months and plants were daily checked in order to sample for flowers, as flowers last for only 24 h.

### 2.4. Flower Traits Obtaining

Every time a plant flowered, the flowers were promptly taken to the laboratory (Figure 3). We assigned a number to each flower in an inflorescence on a scale from one to seven (the maximum number of flowers found in a single inflorescence), where one represented the lowest flower, and subsequent numbers represented higher positions in the inflorescence. In the laboratory, each flower was dried to constant weight, and the biomass (g) was determined in a precision balance (BEL, S2202H, Piracicaba, Brazil).

We also determined each flower’s corolla length, banner petals, nectar guides (considered as flower visibility traits [55]), short and long stamens, and style (considered as primary sexual traits [56]) using a digital caliper (Mtx, Kiev, Ukraine) with 0.001 cm precision.

### 2.5. Data Analysis

All variables were tested for normality with Shapiro–Wilk tests. In order to evaluate the influence of flower position on *E. crassipes* floral traits and its possible interactions with clonal integration, we applied generalized linear mixed models (GLMMs) with Gaussian distribution for all variables, using the package lme4 (version 1.1–35.5) [57]. In order to validate our data distribution choices, we carried out Shapiro–Wilk’s normality tests for the residuals of all models, in which we verified normal distributions for all of them. For every model, the ramet entered the model as a random factor, whereas flower position, treatment (clonal condition, isolated, or defoliation and isolation), and their interaction term were inserted as a fixed variable. After that, we ran a model selection, using the function ‘dredge’, of the package MuMin (version 1.47.5) [58], for each response variable (floral biomass, corolla length, banner petal length, nectar guide length, long stamens length, short stamens length, style length), and the best models were considered to be those composing a group containing the model with the lowest AICc (Akaike Information Criterion corrected for small sample sizes) value and those whose delta (ΔAICc) was lower than 2 [59]. We chose AICc as the sole metric for defining the best models, as it is particularly suited for datasets with small sample sizes and balances model complexity and fit, especially in biologically complex models [60]. The best models were considered to be those with the lowest AICc value or within ΔAICc < 2 [59]. This decision aligns with established recommendations that AIC-based methods are more robust than R^2^ or adjusted R^2^, especially for small datasets, reducing confusion and enhancing reproducibility [61]. The variables were discussed according to their appearance in the candidate models, and their effect direction was discussed based on the summary of the model with the lower AICc among the candidate models, if there were more than one. All analyses were carried out on the R environment [62].

## 3. Results

Flower position emerged as a potential predictor of intra-individual floral trait variation across all determined traits. However, trait responses did not show a consistent pattern. Clonal integration affected floral traits, with significant interactions between flower position and clonal integration observed only for flower visibility traits (Table 1).

Regarding primary sexual functions, defined as the lengths of the long and short stamens and styles, the flower position and treatment terms in the models showed significant influences, although their interaction was not significant (Table 1). For these three traits, the candidate models showed that flower position had a negative effect, indicating that flowers tend to produce shorter structures on apical flowers when compared to the basal ones. However, the best model included only treatment as a significant factor, indicating that resource depletion strongly influenced the structure size and overriding flower position effects (Table 2). An interesting pattern was observed: the ‘Defoliation’ treatment resulted in larger floral structures, except for short stamen length, compared to the ‘Clonal’ treatment. Flowers in the ‘Isolation’ treatment generally exhibited shorter structures than those in the ‘Clonal’ treatment (Table 2).

In relation to flower visibility and pollinator attraction, here listed as floral biomass, corolla length, banner petal length, and nectar guideline length, the interaction term was significant in all of the best models (Table 1). All the determined floral traits, with the exception of floral biomass, decreased in the function of floral position from the bottom to the top of the inflorescence, and this decrease was stronger in plants of the ‘Defoliation’ treatment (Table 2). Nectar guideline length was the only floral trait in the ‘Defoliation’ treatment plants that showed a smoother decrease, while plants in the ‘Isolated’ treatment exhibited the strongest reduction in nectar guideline length as flower position increased (Table 2).

## 4. Discussion

Our results revealed clear patterns of intra-individual variation in floral traits, strongly driven by treatments. For traits associated with primary sexual function, such as stamen and style size, flower position had minimal influence, while resource availability was a key driver for these traits. In the case of traits of floral visibility, associated with pollinator attraction, a combination of flower position and resource availability was the main driver for trait size variation.

In our experiment, the ramets under the lowest resource availability, ‘Defoliation’ (as those plants were split of their parental ramets, precluding resources sharing via clonal integration, and had their leaves removed), exhibited little influence of intra-individual investment on sexual reproductive organs size, which did not happen to floral visibility traits. In sexual reproduction, for instance, individuals allocate resources among traits such as pollinator attraction, floral stalks, and reproductive organs, which would generate a hierarchical resource allocation [63]. This could represent an evolutionary response to environmental stress. For instance, Evans et al. [64] explored physiological integration in *Hydrocotyle bonariensis* under grazing stress, demonstrating that integration can help mitigate resource scarcity effects. Some disturbances may promote sexual reproduction in aquatic plants [65], and under stress conditions, aquatic plants usually allocate a great amount of resources to sexual reproduction when compared to asexual reproduction [66]. 

Severed ramets can, in some cases, produce more inflorescences under nutrient and water limitations, as observed by Evans [67] in *Hydrocotyle bonariensis.* In our experiment, *E. crassipes* ramets may have perceived defoliation and an absence of clonal integration as a high-stress level, which may have biased the decision towards investment in sexual reproductive organs.

In addition to the observed response to defoliation under the absence of clonal integration, it is important to consider the broader ecological implications of resource allocation trade-offs in clonal plants. Clonal plants are often faced with the challenge of balancing investment in vegetative versus reproductive growth, especially under varying environmental conditions [68]. These trade-offs are particularly important because clonal integration can influence how resources are allocated between these competing demands. While some studies suggest that clonal integration can buffer against environmental stress by redistributing resources within the clonal network [67], it may also enable the optimization of reproductive traits in response to specific ecological pressures, such as pollinator availability [69]. On one hand, trade-offs may allow clonal plants to allocate resources flexibly depending on environmental constraints, such as nutrient and water availability [70]. On the other hand, maintaining reproductive investment, especially through traits that attract pollinators, might confer a significant advantage in terms of fitness [69]. In *Eichhornia crassipes*, for example, the ability of ramets to invest in sexual reproduction under stress, such as through increased investment in floral traits, may be a strategy to ensure continued reproduction in the face of fluctuating resource conditions [53]. Moreover, research has shown that traits optimized for reproductive success, such as those involved in pollinator attraction, might be subject to continual selection pressures [71]. This selection could fine-tune floral traits not only to meet the demands of pollinators but also to maximize reproductive success in the context of the plant’s overall life history strategy. These interactions between resource allocation and pollinator-mediated selection further highlight the complexity of trade-offs in clonal plants and their potential for adaptation in dynamic environments.

In *Eichhornia crassipes*, clonal integration has been shown to alleviate trade-offs in vegetative biomass allocation [72], and we suggest the same process may hold true for floral traits. Floral trait stability may be advantageous for attracting consistent pollinator visits, aligning with studies showing that pollinator-mediated selection can drive trait uniformity. Continuous selection of floral traits by pollinators maintains trait integrity, potentially enhancing reproductive success in environments with varying resource availability [73]. A recent study on floral traits integration of *Lonicera* spp. shows that specific floral traits, such as corolla tube length, respond to pollinator preferences [74]. In this sense, certain traits may be optimized for reproductive success under clonal integration, providing evidence that integration might not only buffer stress but also fine-tune traits for environmental or biotic factors like pollinator interactions.

In this sense, why did the ramets of *E. crassipes* in our study not invest in floral visibility, the main driver of pollinator attraction? Besides the apparent importance of pollinator foraging to extensive clonal heterostylous species [75], a previous study carried out in a wide range of *E. crassipes* found a low number of populations showing insect visitation [48], and clones of *Eichhornia crassipes* ramets are reported in the literature as possessing weak or absent self-incompatibility [47] that promotes the production of seeds even under pollinator absence. This selfing behavior is strong in some Brazilian *E. crassipes* populations, in which self- and cross-pollinated plants do not appear to show fertility differences [47]. A high occurrence of selfing was also registered for a population of *E. crassipes* occurring on a seasonal marsh, in which seeds, resistant propagules, are formed right before the dry season [75]. Cao et al. [46] also showed an increase in sexual reproductive structures (ovule and pollen grains) within defoliated ramets. In this way, for the plants in our experiment, ensuring the production of functional reproductive organs appears to be prioritized over flower visibility, which would enhance pollinator attraction, as seeds can be set even in the absence of pollinators.

‘Isolated’ ramets produced smaller structures when compared to ‘Clonal’ and ‘Defoliated’ ramets, and flower position affected floral traits, causing a decrease from the bottom to the top of the inflorescence. Basal flowers can present early anthesis processes and present higher fruit sets in comparison to more distal ones [76], and flower maturity seems to be an important driver of resource investment within inflorescences [77]. However, our data reinforce clonal integration as a pivotal mechanism for clonal plants functioning [38,78] as it enables physiological integration via the maintenance of connections [79]. Our results suggest that clonal integration helps buffer against resource variation, allowing floral traits to remain stable. Clonal integration promotes resilience across ramets in heterogeneous environments, effectively distributing resources to support consistent trait expression [38]. These complex patterns of translocation [79,80], may promote an increase in resource foraging [36], and even a labor division [38,40,42]. In the case of the ‘Isolated’ plants, a ramet bearing a limited amount of roots and leaves was entirely responsible for the nutrient uptake, photosynthesis, and allocation of resources to sexual and asexual reproduction. In the absence of clonal integration with parental plants, the amount of resources available for these functions in each daughter ramet decreased, generating the pattern of flower shortening that was observed along the inflorescence axis that occurred in a smoother way in ‘Clonal’ ramets.

## 5. Conclusions

We conclude that floral traits show significant intra-individual variation in E. crassipes ramets and that this variation is influenced by plant architecture, flower position, resource availability, and the functional nature of the traits. Traits that are linked to the ultimate output of a function, such as stamens and style in relation to sexual reproduction in high fertile self-pollinated plants, suffer less variation than those that are linked to secondary functions, such as pollinator attractiveness, in this case. Our data support the view that sexual reproduction in *E. crassipes* is strongly influenced by environmental conditions [45,46], suggesting that *E. crassipes* populations may not be limited by the absence of pollinators.

## Figures and Tables

**Figure 1 biology-14-00114-f001:**
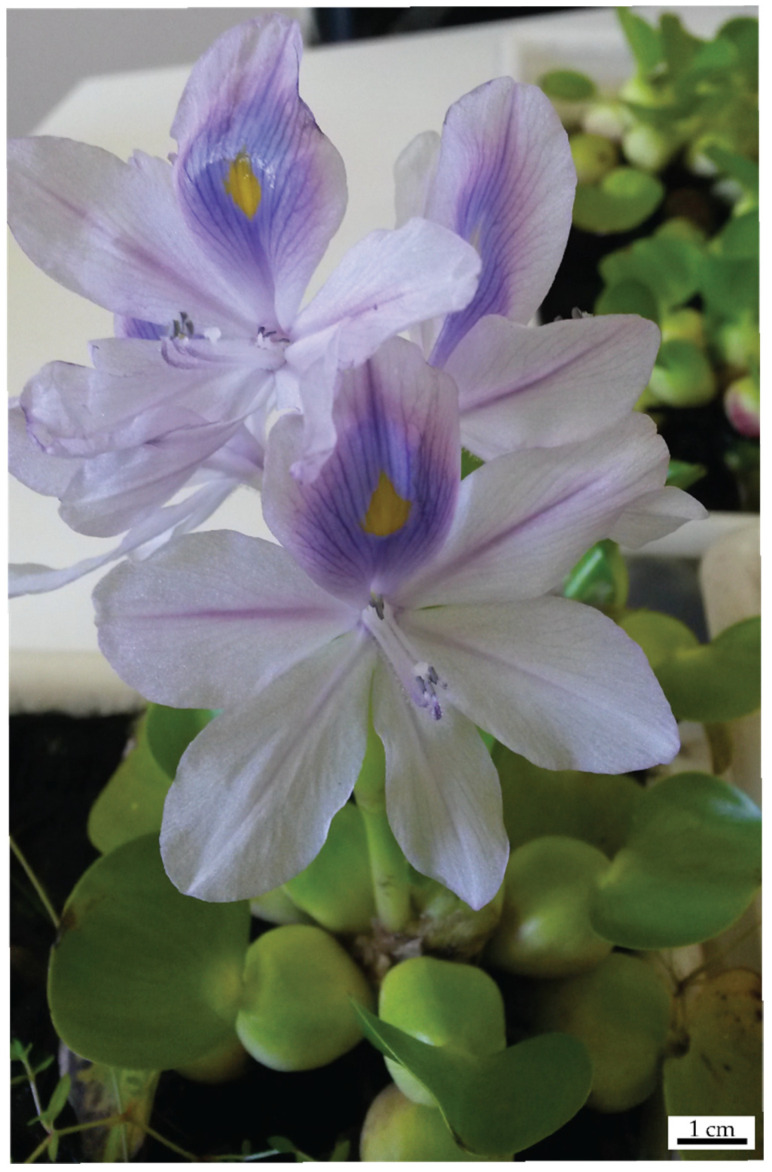
*Eichhornia crassipes* ramet bearing an inflorescence with open flowers.

**Figure 2 biology-14-00114-f002:**
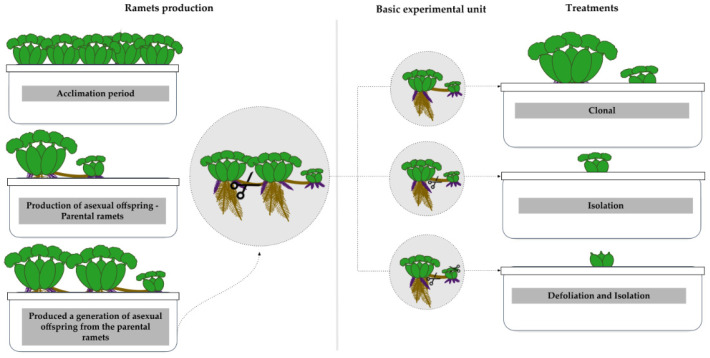
Representative scheme of the experimental steps conducted with *Eichhornia crassipes* ramets in a greenhouse. The scheme illustrates the selection and individual cultivation of ramets until the production of parental ramets, the formation of basic experimental units, and the establishment of the three experimental treatments (clonal, isolation, and defoliation/isolation).

**Figure 3 biology-14-00114-f003:**
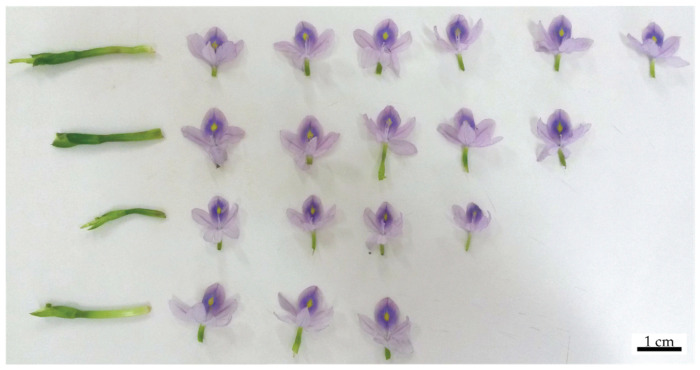
*Eichhornia crassipes* flowers separated in the lab, previously to their traits obtaining. Each line corresponds to the inflorescence of one ramet.

**Table 1 biology-14-00114-t001:** Candidate models for each relationship among *Eichhornia crassipes* floral traits and explanatory variables (flower position, treatment, and interaction term) showing the variables present in each model. A + sign indicates that the term was present in the candidate model.

Trait	Model	Intercept	Flower Position	Treatment	Flower Position × Treatment	AICc	ΔAICc	Weight
Floral biomass	1	0.01758	−0.0002673	+	+	−1118.9	0	0.377
	2	0.01745	−0.0002283	+		−1118.6	0.24	0.334
	3	0.01670		+		−1118.3	0.54	0.228
Corolla length	1	5.554	0.09032	+	+	304.2	0	0.818
Banner petal length	1	3.229	0.01478	+	+	194.6	0	0.996
Nectar guide length	1	0.8407	−0.007613	+	+	62.8	0	0.994
Long stamens length	1	2.216	−0.02395	+		−26.4	0	0.688
Short stamens length	1	0.9371		+		65.3	0	0.56
	2	0.9975	−0.01874	+		66.1	0.76	0.381
Style length	1	1.794		+		160.2	0	0.54
	2	1.716	0.02441	+		161.3	1.09	0.313

**Table 2 biology-14-00114-t002:** Summary coefficients of each of the best models suggested by the model selection regarding the relationships among *Eichhornia crassipes* floral traits and the explanatory variables.

Trait	Component	Estimate	*t* Value	*p*
Floral Biomass	Intercept	0.0175	20.95	**<0.01**
	Treatment (Defoliated)	0.0084	6.947	**<0.01**
	Treatment (Isolated)	−0.011	−10.28	**<0.01**
	Flower Position	−0.0002	−1.252	0.213
	Treatment (Defoliated) × Flower Position	−0.0002	−0.792	0.430
	Treatment (Isolated) × Flower Position	0.0005	1.553	0.123
Corolla length	Intercept	5.554	20.55	**<0.01**
	Treatment (Defoliated)	0.512	1.450	0.149
	Treatment (Isolated)	0.0725	0.217	0.828
	Flower Position	0.0903	1.463	0.146
	Treatment (Defoliated) × Flower Position	−0.261	−2.922	**0.004**
	Treatment (Isolated) × Flower Position	−0.223	−2.159	**0.032**
Banner Petal length	Intercept	3.229	18.53	**<0.01**
	Treatment (Defoliated)	0.5002	2.196	0.030
	Treatment (Isolated)	0.1492	0.692	0.490
	Flower Position	0.0147	0.371	0.711
	Treatment (Defoliated) × Flower Position	−0.2369	−4.097	**<0.01**
	Treatment (Isolated) × Flower Position	−0.0812	−1.214	0.227
Nectar Guide length	Intercept	0.8407	8.246	**<0.01**
	Treatment (Defoliated)	−0.2943	−2.185	**0.030**
	Treatment (Isolated)	0.0381	0.299	0.765
	Flower Position	−0.0076	−0.323	0.747
	Treatment (Defoliated) × Flower Position	0.1569	4.589	**<0.01**
	Treatment (Isolated) × Flower Position	−0.0056	−0.142	0.8876
Long Stamen length	Intercept	2.2162	43.877	**<0.01**
	Treatment (Defoliated)	0.3261	7.242	**<0.001**
	Treatment (Isolated)	−0.2754	−6.174	**<0.01**
	Flower Position	−0.0239	−2.158	**0.032**
Short Stamen length	Intercept	0.9974	10.024	**<0.01**
	Treatment (Defoliated)	−0.1131	−1.794	0.075
	Treatment (Isolated)	−0.4268	−6.912	**<0.01**
	Flower Position	−0.0187	−1.207	0.230
Styles length	Intercept	1.7155	13.023	**<0.01**
	Treatment (Defoliated)	0.2411	2.586	**0.010**
	Treatment (Isolated)	−0.0697	−0.762	0.447
	Flower Position	0.0244	1.062	0.290

Statistically significant *p*-values are presented in bold.

## Data Availability

The raw data supporting the conclusions of this article will be made available by the authors upon request.

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
