# Peer review of "Flower Position and Clonal Integration Drive Intra-Individual Floral Trait Variation in Water-Hyacinth (Eichhornia crassipes, Pontederiaceae)"

_biology, 2025, doi:10.3390/biology14020114_

Round 1
Reviewer 1 Report
Comments and Suggestions for Authors
Article Summary: The article titled “Flower position and clonal integration drive intra-individual floral trait variation in water-lily (Eichhornia crassipes, Pontederiaceae)” investigates intra-individual floral trait variation in E. crassipes under varying environmental conditions and resource availability. The study highlights the influence of flower position, resource allocation, and clonal integration on traits associated with reproductive function and pollinator attractiveness. The main contribution of this article is the demonstration of how clonal integration and flower position within the inflorescence interact to modulate the floral traits of E. crassipes, emphasizing the distinct influences of primary functions (associated with sexual reproduction) and secondary functions (related to pollinator attraction) under varying resource conditions. The article is strong in its use of well-established experimental models but would benefit from certain adjustments. Additionally, the clarity of some technical terms needs improvement, and a review of the text by a specialized company or a native English-speaking professional is recommended.
Review: The review of the topic addressed in the article is well-structured and contributes significantly to the field of ecological and plant physiology. The study on floral trait variation driven by clonal integration and resource availability addresses a clear gap in current knowledge, highlighting its relevance. The references are appropriate, and most of the cited studies complement the theme discussed. However, it would be beneficial to include a more in-depth discussion of some controversial points, such as the ecological implications of resource allocation trade-offs in clonal plants. These could be explored more thoroughly, offering a broader critical perspective on the existing literature.
The manuscript titled " Flower position and clonal integration drive intra-individual floral trait variation in water-lily (Eichhornia crassipes, Pontederiaceae)" presents significant findings. However, to strengthen the data and ensure clarity, the suggested revisions should be fully addressed.
Major revisions
It is recommended that the manuscript is proofread by a specialist company or by a native professional of the English language.
Lines 70 to 72: In the sentence: “In this context, flowers positioned at the base of the inflorescences are prone to receive more resources than those located in more distal positions, acting as a resource sink.” I strongly suggest that the authors provide another explanation (even if theoretical or hypothetical) for the allocation of resources to the flowers positioned at the base of the inflorescences, as there is no purposeless expenditure of energy or resources in biological systems, as mentioned here (resource sink).
Line 77: The term “clonal plants” may cause confusion, particularly for audiences unfamiliar with ecological terminology. Therefore, I recommend briefly defining or conceptualizing the term to enhance understanding.
Line 115: The term “monospecific mat” may cause confusion, particularly for audiences unfamiliar with ecological terminology. Therefore, I recommend briefly defining or conceptualizing the term to enhance understanding.
Minor revisions
Lines 48 and 50: Delete both commas: evolutionary processes“,” and plant fitness [7-8] “,”.
Lines 51 to 53: Rewrite the sentence for clarity: “This intraspecific variation is of main importance for plants, as this variation is ubiquitous in these organisms, and most of the traits that vary in this scale have deep consequences for plant fitness [10].” to “This intraspecific variation is particularly important for plants because it is ubiquitous in these organisms, and many traits that vary at this scale have significant impacts on plant fitness [10].”
Lines 67 to 70: Rewrite the sentence for clarity: “The internal resource gradients caused by proximal-to-distal decline in vasculature size [29-30], specially on plants that show acropetal flowering [30-32] as Eichhornia crassipes, may result in a resource gradient.” to “The internal distribution of resources, driven by the proximal-to-distal reduction in vascular size [29-30], is particularly evident in plants with acropetal flowering patterns [30-32], such as Eichhornia crassipes, and may lead to the formation of a resource gradient.”
Line 89: The term "patches" is widely used in ecology to describe heterogeneous areas in the environment, such as parcels, regions, or zones with greater availability of resources. However, this term may be inappropriate, particularly when the target audience is not familiar with ecological terminology. Therefore, I recommend replacing “patches” with “regions” or “zones” for better clarity and accessibility.
Lines 96 to 98: Rewrite the sentence for clarity: “defoliated ramets should present the highest floral position influence on sexual reproductive traits because flowers located in more proximal positions on the inflorescence would represent a strong resource drain [41-43].” to “defoliated ramets should present greatest influence of floral position on sexual reproductive traits, as flowers in more proximal positions on the inflorescence would act as a strong resource drain [41-43].”
Line 107: Rewrite the sentence for clarity: “pointed” to “highlighted”
Lines 108 to 111: Rewrite the sentence for clarity: “Sexual reproductive structures appear grouped in an inflorescence that generally arises from the apical meristem and bears showy light purple flowers that open in an acropetal order (from the bottom to the top of the inflorescence) (Figure 1).” to “Sexual reproductive structures are grouped in an inflorescence that generally arises from the apical meristem, bearing showy light purple flowers that open in an acropetal order (from bottom to top of the inflorescence), as can be seen in Figure 1.”
Line 115: If possible, add to a periodicity estimate, after “Every time a plant flowered”
Line 131: Delete both commas: offspring “,”
Line 147: Rewrite the sentence for clarity: “2.4. Flower traits measurement” to “2.4. Flower traits obtaining”
Lines 149 to 151: Rewrite the sentence for clarity: “We assigned a number for each flower in an inflorescence, in a scale from one to seven (the maximum number of flowers we found in a unique inflorescence), in what one represented the most bottom flower and the subsequent numbers represented the upper positions in the inflorescence.” to “We assigned a number to each flower in an inflorescence on a scale from one to seven (the maximum number of flowers found in a single inflorescence), where one represented the lowest flower, and subsequent numbers represented higher positions in the inflorescence.”
Line 153: Rewrite the sentence for clarity: “and biomass (g) was measured in a precision balance.” to “and biomass (g) was determined in a precision balance.”
Lines 153 to 157: Rewrite the sentence for clarity: “We also measured each flower’s corolla length, banner petal length, and nectar guideline length (considered as a flower visibility traits [52]), short and long stamens length, and stylus length (considered as primary sexual traits [53]) with a digital caliper with 0.001 cm precision for all flowers in each treatment.” to “We also measured each flower's corolla length, banner petals, nectar guides (considered as flower visibility traits [52]), short and long stamens, and style (considered as primary sexual traits [53]) using a digital caliper with 0.001 cm precision.”
Line 164: Rewrite the sentence for clarity: “GLMM’s” to “GLMMs”
Lines 165 and 171: Please specify the lme4 and MuMin package versions used
Line 174: Add the meaning of this acronym for clarity e.g.: adjusted version of AIC - Akaike Information Criterion (AICc)
Lines 187 to 189: Rewrite the sentence for clarity: “Regarding sexual primary function, here listed as long stamens length, short stamens length, and style length, flower position and treatment terms of the models showed influences, with the interaction being not significant (Table 1).” to “Regarding primary sexual functions, defined as the lengths of the long and short stamens and styles, the flower position and treatment terms in the models showed significant influences, although their interaction was not significant (Table 1).”
Lines 187 to 189: “Here, an interesting pattern arose.” Is a very informal phrase; I suggest this be rephrased into something more technical like: “An interesting pattern was observed.”
Lines 194 to 197: Rewrite the sentence for clarity: “The ‘Defoliation’ treatment resulted in larger floral structures, except for the short stamen length than the ‘Clonal’ treatment, while flowers in the ‘Isolation’ treatment generally exhibited shorter structures than those in the ‘Clonal’ treatment (Table 2).” to “The ‘Defoliation’ treatment resulted in larger floral structures, except for short stamen length, compared to the ‘Clonal’ treatment. Flowers in the ‘Isolation’ treatment generally exhibited shorter structures than those in the ‘Clonal’ treatment (Table 2).”
Line 202: “measured” to “determined”
Lines 205 to 208: Rewrite the sentence for clarity: “Nectar guideline length was the only floral trait of ‘Defoliation’ treatment plants that had a smoother decrease in relation to the treatments, with ‘Isolated’ treatment plants showing the strongest reduction in nectar guidelines length with the increase of flower position (Table 2). ” to “Nectar guideline length was the only floral trait in the ‘Defoliation’ treatment plants that showed a smoother decrease, while plants in the ‘Isolated’ treatment exhibited the strongest reduction in nectar guideline length as flower position increased (Table 2).”
Line 233: Rewrite the sentence for clarity: “Eicchornia” to “Eichhornia”
Lines 278 to 280: Rewrite the sentence for clarity: “We conclude that floral traits show great intra-individual variation in E. crassipes ramets, and that this variation is driven by architecture, mediated by flower position, resource availability, and by the nature of the trait function.” to “We conclude that floral traits show significant intra-individual variation in E. crassipes ramets, and that this variation is influenced by plant architecture, flower position, resource availability, and the functional nature of the traits.”
Line 285: Rewrite the sentence for clarity: “Water-lily” is a more generic term and may cause confusion. Replacing it with “E. crassipes populations” would provide greater accuracy.
Comments on the Quality of English LanguageIt is recommended that the manuscript is proofread by a specialist company or by a native professional of the English language.
Author Response
Article Summary: The article titled “Flower position and clonal integration drive intra-individual floral trait variation in water-lily (Eichhornia crassipes, Pontederiaceae)” investigates intra-individual floral trait variation in E. crassipes under varying environmental conditions and resource availability. The study highlights the influence of flower position, resource allocation, and clonal integration on traits associated with reproductive function and pollinator attractiveness. The main contribution of this article is the demonstration of how clonal integration and flower position within the inflorescence interact to modulate the floral traits of E. crassipes, emphasizing the distinct influences of primary functions (associated with sexual reproduction) and secondary functions (related to pollinator attraction) under varying resource conditions. The article is strong in its use of well-established experimental models but would benefit from certain adjustments. Additionally, the clarity of some technical terms needs improvement, and a review of the text by a specialized company or a native English-speaking professional is recommended.
Review: The review of the topic addressed in the article is well-structured and contributes significantly to the field of ecological and plant physiology. The study on floral trait variation driven by clonal integration and resource availability addresses a clear gap in current knowledge, highlighting its relevance. The references are appropriate, and most of the cited studies complement the theme discussed. However, it would be beneficial to include a more in-depth discussion of some controversial points, such as the ecological implications of resource allocation trade-offs in clonal plants. These could be explored more thoroughly, offering a broader critical perspective on the existing literature.
The manuscript titled " Flower position and clonal integration drive intra-individual floral trait variation in water-lily (Eichhornia crassipes, Pontederiaceae)" presents significant findings. However, to strengthen the data and ensure clarity, the suggested revisions should be fully addressed.
Thank you for the thoughtful summary and detailed feedback on our manuscript. We are pleased to know that you found our study well-structured, relevant, and a valuable contribution to the field of ecological and plant physiology.
- Clarity of Technical Terms:
We appreciate your recommendation to enhance the clarity of technical terms. In response, we have carefully reviewed the manuscript and revised sections where terms may have been unclear, ensuring they are accessible to a broader audience. Additionally, we have revised the English to refine the overall readability and precision of the text. - Ecological Implications of Resource Allocation Trade-Offs:
We agree that a deeper discussion of the ecological implications of resource allocation trade-offs in clonal plants would strengthen the manuscript. We have expanded the Discussion section (Lines 262-285) to address these trade-offs, emphasizing how resource allocation patterns in clonal plants, such as Eichhornia crassipes, influence ecological interactions and reproductive strategies. This addition provides a broader critical perspective on the literature and situates our findings within a larger ecological context. - Suggested Revisions:
We have carefully addressed all suggested revisions, including ensuring clarity and providing the requested adjustments to strengthen our data presentation. Specific changes include the addition of scale bars in Figures 1 and 3.
We hope these revisions address your concerns and enhance the clarity and impact of our manuscript. All the changes referring to this revision are marked in yellow. Those marked in blue refer to the comments made by reviewer 2. Thank you again for your valuable input, which has greatly contributed to improving the quality of this work.
Major revisions
It is recommended that the manuscript is proofread by a specialist company or by a native professional of the English language.
Thank you for your recommendation regarding the language quality of the manuscript. To ensure clarity and precision, we have arranged for the manuscript to be reviewed and proofread by a native English-speaking professional specializing in scientific writing. We believe this will significantly enhance the readability and overall quality of the manuscript.
Lines 70 to 72: In the sentence: “In this context, flowers positioned at the base of the inflorescences are prone to receive more resources than those located in more distal positions, acting as a resource sink.” I strongly suggest that the authors provide another explanation (even if theoretical or hypothetical) for the allocation of resources to the flowers positioned at the base of the inflorescences, as there is no purposeless expenditure of energy or resources in biological systems, as mentioned here (resource sink).
Thank you for your insightful suggestion regarding the explanation of resource allocation to flowers positioned at the base of inflorescences. We agree that the concept of a "resource sink" warrants further clarification, and we have revised the sentence accordingly to reflect the biological purpose of resource allocation. In the revised manuscript, we have incorporated the source-sink dynamics of plant organs to explain this phenomenon. Flowers at the base of inflorescences are typically among the first to develop and reach anthesis, placing them in a favorable position to act as primary sinks for resources translocated from source organs (e.g., leaves). This pattern aligns with the developmental priority hypothesis, wherein proximal flowers initiate earlier and attract more resources due to their early establishment as active sinks. These resources are allocated strategically to maximize reproductive success, ensuring that basal flowers, which are more likely to set fruit due to their proximity to the vascular supply, receive preferential investment. We believe this revised explanation (Lines 69-83) strengthens the biological rationale for resource allocation patterns and aligns with established principles of plant physiology. Thank you again for raising this important point, which has allowed us to improve the clarity and scientific grounding of our manuscript.
Line 77: The term “clonal plants” may cause confusion, particularly for audiences unfamiliar with ecological terminology. Therefore, I recommend briefly defining or conceptualizing the term to enhance understanding.
Thank you for your valuable feedback. We agree that the term 'clonal plants' may be unfamiliar to some readers, and we will add a brief definition to enhance understanding. We included the following conceptualization in the manuscript: 'Clonal plants are species that reproduce asexually through vegetative structures, such as runners, stolons, or rhizomes, allowing them to form genetically identical offspring or clones, which, in most cases, remain attached to their parental plants (Lines 88-91).' This should provide a clearer understanding for the audience.
Line 115: The term “monospecific mat” may cause confusion, particularly for audiences unfamiliar with ecological terminology. Therefore, I recommend briefly defining or conceptualizing the term to enhance understanding.
Thank you for your helpful suggestion. We understand that the term 'monospecific mat' might be unfamiliar to some readers, and we appreciate your recommendation to clarify its meaning. We included the following definition in the manuscript: 'a large, continuous patch of clonal plants composed of a single species (E. crassipes in this study), where vegetative offspring (e.g., generated by runners, rhizomes, or stolons) spread from a central parent plant to form a dense network of genetically identical individuals.' (Lines 130-133). This should help improve the clarity of the term for the audience.
Minor revisions
Lines 48 and 50: Delete both commas: evolutionary processes“,” and plant fitness [7-8] “,”.
Thank you for your feedback. We have deleted the commas as suggested (Lines 49-52). We appreciate your attention to detail.
Lines 51 to 53: Rewrite the sentence for clarity: “This intraspecific variation is of main importance for plants, as this variation is ubiquitous in these organisms, and most of the traits that vary in this scale have deep consequences for plant fitness [10].” to “This intraspecific variation is particularly important for plants because it is ubiquitous in these organisms, and many traits that vary at this scale have significant impacts on plant fitness [10].”
Thank you for your suggestion. We have revised the sentence as requested (Lines 52-55). We appreciate your help in improving the clarity of the text.
Lines 67 to 70: Rewrite the sentence for clarity: “The internal resource gradients caused by proximal-to-distal decline in vasculature size [29-30], specially on plants that show acropetal flowering [30-32] as Eichhornia crassipes, may result in a resource gradient.” to “The internal distribution of resources, driven by the proximal-to-distal reduction in vascular size [29-30], is particularly evident in plants with acropetal flowering patterns [30-32], such as Eichhornia crassipes, and may lead to the formation of a resource gradient.”
Thank you for your suggestion. We have revised the sentence as requested. (Lines 70-73). We appreciate your help in improving the clarity of the text.
Line 89: The term "patches" is widely used in ecology to describe heterogeneous areas in the environment, such as parcels, regions, or zones with greater availability of resources. However, this term may be inappropriate, particularly when the target audience is not familiar with ecological terminology. Therefore, I recommend replacing “patches” with “regions” or “zones” for better clarity and accessibility.
Thank you for your insightful comment. We have replaced the term 'patches' with 'area' (Line 89) to enhance clarity and accessibility, as recommended. The revised text now reads 'resource-rich areas'. We appreciate your suggestion to improve the understanding of the term.
Lines 96 to 98: Rewrite the sentence for clarity: “defoliated ramets should present the highest floral position influence on sexual reproductive traits because flowers located in more proximal positions on the inflorescence would represent a strong resource drain [41-43].” to “defoliated ramets should present greatest influence of floral position on sexual reproductive traits, as flowers in more proximal positions on the inflorescence would act as a strong resource drain [41-43].”
Thank you for your suggestion. We have revised the sentence as requested (Lines 111-113). We appreciate your help in improving the clarity of the text.
Line 107: Rewrite the sentence for clarity: “pointed” to “highlighted”
Thank you for your suggestion. We have replaced 'pointed' with 'highlighted' (Line 122) as recommended. The revised sentence now reflects the change for improved clarity.
Lines 108 to 111: Rewrite the sentence for clarity: “Sexual reproductive structures appear grouped in an inflorescence that generally arises from the apical meristem and bears showy light purple flowers that open in an acropetal order (from the bottom to the top of the inflorescence) (Figure 1).” to “Sexual reproductive structures are grouped in an inflorescence that generally arises from the apical meristem, bearing showy light purple flowers that open in an acropetal order (from bottom to top of the inflorescence), as can be seen in Figure 1.”
Thank you for your suggestion. We have revised the sentence as requested (lines 123-126). We appreciate your help in improving the clarity of the text.
Line 115: If possible, add to a periodicity estimate, after “Every time a plant flowered”
Thank you for your suggestion. While we understand the importance of specifying periodicity, it is important to note that the different treatments flowered with varying periodicities. A mean periodicity would not accurately represent the actual flowering intervals of E. crassipes. Due to this variability, we chose not to include a time measure for periodicity in our study. Instead, we focused on collecting flowers promptly every time they flowered, as described in the text (line 168). We hope this clarifies our approach and reasoning.
Line 131: Delete both commas: offspring “,”
Thank you for your suggestion. We have revised the sentence as requested (line 151). We appreciate your help in improving the clarity of the text.
Line 147: Rewrite the sentence for clarity: “2.4. Flower traits measurement” to “2.4. Flower traits obtaining”
Thank you for your suggestion. We have revised the sentence as requested (line 167). We appreciate your help in improving the clarity of the text.
Lines 149 to 151: Rewrite the sentence for clarity: “We assigned a number for each flower in an inflorescence, in a scale from one to seven (the maximum number of flowers we found in a unique inflorescence), in what one represented the most bottom flower and the subsequent numbers represented the upper positions in the inflorescence.” to “We assigned a number to each flower in an inflorescence on a scale from one to seven (the maximum number of flowers found in a single inflorescence), where one represented the lowest flower, and subsequent numbers represented higher positions in the inflorescence.”
Thank you for your suggestion. We have revised the sentence as requested (Lines 169-172). We appreciate your help in improving the clarity of the text.
Line 153: Rewrite the sentence for clarity: “and biomass (g) was measured in a precision balance.” to “and biomass (g) was determined in a precision balance.”
Thank you for your suggestion. We have revised the sentence as requested (Line 173). We appreciate your help in improving the clarity of the text.
Lines 153 to 157: Rewrite the sentence for clarity: “We also measured each flower’s corolla length, banner petal length, and nectar guideline length (considered as a flower visibility traits [52]), short and long stamens length, and stylus length (considered as primary sexual traits [53]) with a digital caliper with 0.001 cm precision for all flowers in each treatment.” to “We also measured each flower's corolla length, banner petals, nectar guides (considered as flower visibility traits [52]), short and long stamens, and style (considered as primary sexual traits [53]) using a digital caliper with 0.001 cm precision.”
Thank you for your suggestion. We have revised the sentence as requested (Lines 173-176). We appreciate your help in improving the clarity of the text.
Line 164: Rewrite the sentence for clarity: “GLMM’s” to “GLMMs”
Thank you for your suggestion. We have revised the sentence as requested (Line 184). We appreciate your help in improving the clarity of the text.
Lines 165 and 171: Please specify the lme4 and MuMin package versions used
Thank you for your suggestion. We have specified the versions of the lme4 (Lines 184-185) and MuMin (Line 191) packages used in the manuscript, as requested.
Line 174: Add the meaning of this acronym for clarity e.g.: adjusted version of AIC - Akaike Information Criterion (AICc)
Thank you for your helpful suggestion. We have added the meaning of the acronym as recommended: 'Akaike Information Criterion corrected for small sample sizes' (Lines 194-195). This should provide greater clarity for the readers.
Lines 187 to 189: Rewrite the sentence for clarity: “Regarding sexual primary function, here listed as long stamens length, short stamens length, and style length, flower position and treatment terms of the models showed influences, with the interaction being not significant (Table 1).” to “Regarding primary sexual functions, defined as the lengths of the long and short stamens and styles, the flower position and treatment terms in the models showed significant influences, although their interaction was not significant (Table 1).
Thank you for your suggestion. We have revised the sentence as requested (lines 215-217). We appreciate your help in improving the clarity of the text.
Lines 187 to 189: “Here, an interesting pattern arose.” Is a very informal phrase; I suggest this be rephrased into something more technical like: “An interesting pattern was observed.”
Thank you for your suggestion. We have revised the sentence as requested (line 222). We appreciate your help in improving the clarity of the text.
Lines 194 to 197: Rewrite the sentence for clarity: “The ‘Defoliation’ treatment resulted in larger floral structures, except for the short stamen length than the ‘Clonal’ treatment, while flowers in the ‘Isolation’ treatment generally exhibited shorter structures than those in the ‘Clonal’ treatment (Table 2).” to “The ‘Defoliation’ treatment resulted in larger floral structures, except for short stamen length, compared to the ‘Clonal’ treatment. Flowers in the ‘Isolation’ treatment generally exhibited shorter structures than those in the ‘Clonal’ treatment (Table 2).”
Thank you for your suggestion. We have revised the sentence as requested (lines 222-225). We appreciate your help in improving the clarity of the text.
Line 202: “measured” to “determined”
Thank you for your suggestion. We have replaced 'measured' with 'determined' throughout the manuscript to standardize the terminology, as requested.
Lines 205 to 208: Rewrite the sentence for clarity: “Nectar guideline length was the only floral trait of ‘Defoliation’ treatment plants that had a smoother decrease in relation to the treatments, with ‘Isolated’ treatment plants showing the strongest reduction in nectar guidelines length with the increase of flower position (Table 2). ” to “Nectar guideline length was the only floral trait in the ‘Defoliation’ treatment plants that showed a smoother decrease, while plants in the ‘Isolated’ treatment exhibited the strongest reduction in nectar guideline length as flower position increased (Table 2).”
Thank you for your suggestion. We have revised the sentence as requested (lines 233-236) We appreciate your help in improving the clarity of the text.
Line 233: Rewrite the sentence for clarity: “Eicchornia” to “Eichhornia”
Thank you for pointing that out. We have corrected the species name from 'Eicchornia' to 'Eichhornia' and have carefully searched the text to ensure it is correctly written throughout."
Lines 278 to 280: Rewrite the sentence for clarity: “We conclude that floral traits show great intra-individual variation in E. crassipes ramets, and that this variation is driven by architecture, mediated by flower position, resource availability, and by the nature of the trait function.” to “We conclude that floral traits show significant intra-individual variation in E. crassipes ramets, and that this variation is influenced by plant architecture, flower position, resource availability, and the functional nature of the traits.”
Thank you for your suggestion. We have revised the sentence as requested (lines 333-335). We appreciate your help in improving the clarity of the text.
Line 285: Rewrite the sentence for clarity: “Water-lily” is a more generic term and may cause confusion. Replacing it with “E. crassipes populations” would provide greater accuracy.
Thank you for your suggestion. We have revised the sentence as requested (Line 340). We appreciate your help in improving the clarity of the text.
Reviewer 2 Report
Comments and Suggestions for Authors
The authors described the water hyacinth (Eichhornia crassipes), a unique clonally growing invasive aquatic plant, and explores how flower position and clonal integration affect intra-individual variation in floral traits. The paper links clonal integration to floral trait stability provides a new perspective for understanding how plants adapt to variable environments.
1. Lines 112 and 158, there was missing bar in the Figures 1 and 3.
2. Line 164, the paper used Generalized Linear Mixed Models (GLMMs) to analyze the effects of flower position and clonal integration on floral traits and employs model selection to determine the best models. This approach is suitable for handling data with hierarchical structures and random effects, effectively reflecting the complex relationships in the experimental design. However, the model selection process relies solely on AICc values without considering other criteria (e.g., BIC, adjusted R²), which may limit the optimality and interpretability of the models. It would be better to incorporate multiple model selection criteria for a more comprehensive analysis.
Comments on the Quality of English Language
The English could be improved to more clearly express the research.
Author Response
The authors described the water hyacinth (Eichhornia crassipes), a unique clonally growing invasive aquatic plant, and explores how flower position and clonal integration affect intra-individual variation in floral traits. The paper links clonal integration to floral trait stability provides a new perspective for understanding how plants adapt to variable environments.
Thank you for your thoughtful and positive comment. We are pleased that you found our discussion on Eichhornia crassipes and its unique clonal growth strategy compelling. Indeed, our work aimed to explore how flower position and clonal integration influence intra-individual variation in floral traits, shedding light on the relationship between clonal integration and floral trait stability. By linking these dynamics, we sought to contribute to a broader understanding of how clonal plants adapt to spatially and temporally variable environments. We appreciate that you recognized this novel perspective, and we hope our findings will inspire further research into the ecological and evolutionary implications of clonal integration in plant populations.
- Lines 112 and 158, there was missing bar in the Figures 1 and 3.
Thank you for pointing out the missing scale bars in Figures 1 and 3. We have now added the scale bars to these figures for clarity and accuracy.
- Line 164, the paper used Generalized Linear Mixed Models (GLMMs) to analyze the effects of flower position and clonal integration on floral traits and employs model selection to determine the best models. This approach is suitable for handling data with hierarchical structures and random effects, effectively reflecting the complex relationships in the experimental design. However, the model selection process relies solely on AICc values without considering other criteria (e.g., BIC, adjusted R²), which may limit the optimality and interpretability of the models. It would be better to incorporate multiple model selection criteria for a more comprehensive analysis.
Thank you for your thoughtful and constructive feedback regarding the model selection process. We appreciate your recognition of the suitability of GLMMs for analyzing hierarchical data and reflecting the complex relationships in our experimental design. Regarding the use of AICc as the sole criterion for model selection, we chose it intentionally due to its strengths in evaluating models, particularly when dealing with datasets with a small sample size, as in our study. AICc is widely regarded as a robust metric for balancing model fit and complexity, avoiding overfitting while providing biologically meaningful insights. As Emiliano et al. (2014) (https://doi.org/10.1016/j.csda.2013.07.032) demonstrated, for biological growth model simulations with very small sample sizes, AIC and AICc outperform criteria such as BIC in terms of identifying optimal models. Furthermore, Jenkins and Quintana-Ascencio (2020) (https://doi.org/10.1371/journal.pone.0229345) emphasized that alternative models are better compared using information-theoretic indices like AIC, rather than R2 or adjusted R2. Small sample sizes andR2-based model selection have been associated with confusion and low reproducibility in various disciplines, further supporting our decision to rely on AICc. While additional criteria such as BIC or adjusted R2 could provide complementary perspectives, we aimed to maintain a clear and consistent framework for model evaluation that aligned with our biological hypotheses. Moreover, BIC tends to favor simpler models, which may not fully capture the nuanced effects of clonal integration and flower position on floral traits in our experimental setup. To address your concern, we have revised the manuscript to include a detailed justification for our reliance on AICc, citing its demonstrated performance in similar scenarios and its alignment with our study's objectives. We have also acknowledged the potential value of integrating multiple model selection criteria in future studies to validate findings further. This clarification, now included in the Data analysis (Lines 195-201) section, enhances the transparency of our methodological choices. We hope this response and the accompanying revisions clarify our approach and strengthen the manuscript.
Round 2
Reviewer 2 Report
Comments and Suggestions for Authors Your response to the review comments is quite comprehensive, and the revised manuscript has made significant improvements in many aspects.